Risk of circulatory diseases associated with proton-pump inhibitors: a retrospective cohort study using electronic medical records in Thailand

http://orcid.org/0000-0002-7316-0094 Pannoi Tanavij 1 2 joob103@gmail.com
Promchai Chissanupong 3
Apiromruck Penjamaporn 3
Wongpraphairot Suwikran 4
Dong Yaa-Hui 5 6 7
Yang Chen-Chang 2 8 9
Pan Wen-Chi 2 8 wenchipan@nycu.edu.tw
1 Department of Pharmaceutical Care, School of Pharmacy, Walailak University , Nakhornsrithammarat , Thailand
2 International Health Program, Institute of Public Health, National Yang Ming Chiao Tung University , Taipei , Taiwan
3 Department of Pharmacy, Songklanagarind Hospital, Prince of Songkhla University , Songkhla , Thailand
4 Department of Internal Medicine, Songklanagarind Hospital, Prince of Songkhla University , Songkhla , Thailand
5 Department of Pharmacy, College of Pharmaceutical Sciences, National Yang Ming Chiao Tung University , Taipei , Taiwan
6 Institute of Public Health, College of Medicine, National Yang Ming Chiao Tung University , Taipei , Taiwan
7 Institute of Hospital and Health Care Administration, College of Medicine, National Yang Ming Chiao Tung University , Taipei , Taiwan
8 Institute of Environmental and Occupational Health Sciences, National Yang Ming Chiao Tung University , Taipei , Taiwan
9 Department of Occupational Medicine and Clinical Toxicology, Taipei Veterans General Hospital , Taipei , Taiwan
Yang Guangdong
Electronic publication date: 2024 Feb 15
Publication date: 2024
Volume: 12
Electronic Location ID: e16892
Received 2023 Sep 15; Accepted 2024 Jan 16
Copyright: © 2024 Pannoi et al.
Copyright year: 2024
Copyright holder: Pannoi et al.
License: This is an open access article distributed under the terms of the Creative Commons Attribution License, which permits unrestricted use, distribution, reproduction and adaptation in any medium and for any purpose provided that it is properly attributed. For attribution, the original author(s), title, publication source (PeerJ) and either DOI or URL of the article must be cited.
License URL: https://creativecommons.org/licenses/by/4.0/

Keywords: Proton-pump inhibitors, Circulatory diseases, Cardiovascular diseases, Ischemic strokes, Peripheral vascular diseases, Retrospective cohort, Electronic medical records, Pharmacoepidemiology

Funding: The authors received no funding for this work.

==============================
Background

Proton-pump inhibitors (PPIs) are prescribed to treat gastric acid-related diseases, while they may also have potential risks to population health. Recent studies suggested that a potential mechanism explaining the association between PPIs and cardiovascular diseases (CVD) includes the inhibition of the nitrate-nitrite-nitric oxide (NO) pathway. However, previous observational studies showed controversial results of the association. In addition, the inhibition of the NO pathway due to PPIs use may lead to peripheral vascular diseases (PVD); however, none of the studies explore the PPI-PVD association. Therefore, this study aimed to evaluate the association of PPIs with circulatory diseases (CVD, ischemic strokes or IS, and PVD).

Methods

We conducted a retrospective hospital-based cohort study from Oct 2010 to Sep 2017 in Songkhla province, Thailand. PPIs and histamine 2-receptor antagonists (H2RAs) prescriptions were collected from electronic pharmacy records, while diagnostic outcomes were retrieved from electronic medical records at Songklanagarind hospital. Patients were followed up with an on-treatment approach. Cox proportional hazard models were applied to measure the association comparing PPIs vs H2RAs after 1:1 propensity-score-matching. Sub-group analysis, multi-bias E-values, and array-based sensitivity analysis for some covariates were used to assess the robustness of associations.

Results

A total of 3,928 new PPIs and 3,928 H2RAs users were included in the 1:1 propensity score-matched cohort. As compared with H2RAs, the association of PPIs with CVD, IS, and PVD, the hazard ratios were 1.76 95% CI = [1.40–2.20] for CVD, 3.53 95% CI = [2.21–5.64] for ischemic strokes, and 17.07 95% CI = [13.82–76.25] for PVD. The association between PPIs and each outcome was significant with medication persistent ratio of over 50%. In addition, the association between PPIs and circulatory diseases was robust to unmeasured confounders (i.e., smoking and alcohol).

Conclusion

PPIs were associated with circulatory diseases, particularly ischemic strokes in this hospital-based cohort study, whereas, the strength of associations was robust to unmeasured confounders.

Introduction

Proton-pump inhibitors (PPIs) have become a drug of choice for the treatment of acid-related disorders since Omeprazole was introduced in 1989. Following that, PPIs outperformed the effectiveness of other gastric-acid suppressants (e.g., histamine 2-receptor antagonists (H2RAs)).

In 2010, however, the U.S. Food and Drug Administration (FDA) warned manufacturers to include the risk for bone fractures on the product labels when a high dose or long-term use of PPIs was prescribed (Strand, Kim & Peura, 2017). Since then, many studies reported more potential risks of PPIs use, including renal impairment, infections, and cardiovascular diseases (CVD) (Corsonello & Lattanzio, 2018). A meta-analysis of randomized controlled trials showed that monotherapy of PPIs was associated with 70% increase in cardiovascular diseases (relative risk (RR) = 1.70, 95% Confidence Intervals 95% CI = [1.13–2.56]) (Sun et al., 2017). Likewise, a meta-analysis of observational studies reported that the rates of cardiovascular diseases in PPIs users were significantly higher than in non-PPIs users (Odds ratio (OR) = 1.54, 95% CI = [1.11–2.13]) (Shiraev & Bullen, 2018).

Regarding in vitro studies, the potential mechanism underlying the association between PPIs and CVD was the inhibition of the nitrate-nitrite-nitric oxide (NO) pathway. It was also observed that omeprazole prevented the effects of nitrite and nitrate; the rise of pH in the stomach inhibited NO and other NO-related species synthesis. This consequently diminishes a cardiovascular protective effect (Amaral & Tanus-Santos, 2017; Pinheiro et al., 2016). In addition to a recent in vivo study, the long-term use of PPIs might inhibit dimethylarginine dimethylaminohydrolase (DDAH) activity, impair endothelial NO production, and reduce vascular function (Ghebremariam et al., 2015; Nolde et al., 2021). This mechanism was possibly linked to the impairment of circulatory system, including ischemic strokes (IS) and peripheral vascular diseases (PVD).

Nevertheless, the association between PPIs use and IS remained controversial among large observational studies. For instance, Sehested et al. (2018) conducted a study among all registry-Danish populations who underwent an elective upper gastrointestinal endoscopy during 1997 to 2012, had a 29% (95% CI = [5–59%]) greater absolute risk of ischemic stroke than those people who had not used PPIs. On the contrary, Nguyen et al. (2018) studied sex-different cohorts of health professionals and found non-significant associations (Adjusted Hazard Ratio (aHR) = 1.08; 95% CI = [0.91–1.27]).

Noticeably, the results remained inconclusive because of the different populations included in each study. A recent systematic review and meta-analysis also observed that the Asian population reported a substantially higher effect of PPIs use on the risk of cardiovascular events than studies from other regions. However, there were only two Asian studies, which was not sufficient to decide whether this was a coincidence or an actual effect modification (Nolde et al., 2022). In addition, non-user of PPIs was applied as a comparator in many studies, which led to indication bias on estimated outcomes.

Moreover, the potential mechanism underlying the association between PPIs and impaired vascular function has not yet been studied for PVD as a primary outcome. Although there were systematic review and meta-analysis studies on PPIs use associated with CVD, results remained inconsistent owing to medium to high risk biases among the pooled studies (Nolde et al., 2022; Sherwood et al., 2015; Shiraev & Bullen, 2018).

Therefore, this study aimed to assess the association between PPIs use and circulatory diseases (CVD, IS, and PVD) using electronic medical records. A reproducible observational study of circulatory disease-related to PPIs use in the Asian population is required to address the aforementioned issues, whereas, using the real-world clinical database is to gain more meaningful in clinical practice.

Materials and Methods

Study design

A retrospective cohort study was conducted to compare patients, initiated by PPIs or H2RAs (as an active comparator). We identified all outpatient or inpatient visits, who were firstly prescribed either PPIs or H2RAs from October 1, 2010, to September 30, 2017. Patients were excluded if they had been prescribed either drug for 6 months before the entry year, which was the fiscal year 2011 (October 1, 2010, to September 30, 2011) or 2012 (October 1, 2011, to September 30, 2012) (Fig. 1).

Figure 1 Study design diagram.

PPIs, Proton-pump inhibitors; H2RAs, Histamine-2 receptor antagonist; VD, Circulatory diseases. * Treatment episodes were defined by date of dispensing and day’s supply with a stockpiling algorithm if a new dispensing occurred before the end of days’ supply. Gaps of <90 days between end of days’ supply and next dispensing (grace period) were bridged. 90 days was added to the last dispensing days’ supply in an exposure episode. ŦCensor was due to discontinuation at grace period = 90 days, death, or switched treatment. ±Outcomes were ICD-10 related with circulatory diseases; cardio vascular disease, ischemic strokes, and peripheral vascular disease.

Study population

Patient information was retrieved from medical and administrative databases at Songklanagarind Hospital, Thailand. A description of the study hospital information was provided in the Supplemental materials. The datasets consisted of inpatient, outpatient, medication, laboratory, and administrative data. For inclusion criteria, all patients who were over 20 years old and administered either PPIs or H2RAs during the entry years were recruited (Fig. 2).

Figure 2 Flowchart of participant’s selection.

*CVD, Cardio vascular diseases; **IS, Ischemic strokes; ***PVD, Peripheral vascular diseases.

Ethical consideration

The patient’s informed consent according to the Declaration of Helsinki and the International Conference on Harmonization in Good Clinical Practice was waived by institutional review boards (IRB) of National Yang Ming Chiao Tung University (the approval number is YM108156E). Data collection was permitted by the hospital director about IRB approval from the study site (the approval number is 63-096-19-9).

Exposure measurement

Since the entry period, participants, who were initially given by PPIs or H2RAs, were defined as eligible new users of PPIs or H2RAs, along with the individual index dates. The on-treatment follow-up scheme was applied in this study. Since the discontinuation of medication time was unknown in observational data, we pre-defined the discontinuation time of medication by the last prescription dispensing date plus the days’ supply and grace period. Participants who did not refill a prescription for PPIs or H2RAs before discontinuation were excluded. A 90-day grace period was defined as the main analysis because a steady increase in 3-month prescriptions dispensed was observed in the dataset.

Outcome measurement

The primary outcomes were the incidence of cardiovascular disease (CVD), ischemic strokes (IS), and peripheral vascular disease (PVD) during the observational period, with a maximum of 7 years followed from October 1, 2010 to September 30, 2017. Details on the International Classification of Diseases version 10th (ICD-10) code for outcomes are outlined in the table at supplemental (Table 2S).

Covariates measurement

Participants’ characteristics, including hospital service utilization, medications, and diagnoses, were retrieved and presented as baseline characteristics. Individual blood pressure measures (systolic and diastolic blood pressure) before the index time (T0) were collected based on the availability of the data. Since we used retrospective data, the reasons for missing data could not be reassessed. Therefore, we assumed that probability of being missing was the same for all participants, namely, missing completely at random (MCAR). Due to missing systolic and diastolic blood pressure (SBP and DBP, respectively) data for some participants, multiple imputation with mice package in R based on sex and age was applied for the unobserved participants’ SBP and DBP.

Statistical analysis

Individuals who were diagnosed with primary outcomes before the index date were excluded from analysis. Participants who switched between PPIs and H2RAs treatments, died, or did not develop major outcomes during the follow-up period were defined as having right-censorship (Fig. 1).

The baseline characteristics of PPI and H2RAs users were reported as frequency, percentage, mean, and standard deviation, or median and interquartile range, as appropriate. In addition, the baseline characteristics of the exposure group and the active comparator were compared using the absolute standardized difference (ASD) (Yang & Dalton, 2012) between the two groups for categorical data. We adjusted for covariates in which the ASD was more than 0.10 (Table 1), in each model.

Table 1 Baseline demographic and health characteristics of overall cohort between proton-pump inhibitors (PPIs) and histamine-2-receptor antagonist (H2RAs) users.

Covariates		Before propensity score matching	After 1:1 Propensity score matching	
Overall	H2RAs	PPIs	Absolute standardized differenceŦ	Overall	H2RAs	PPIs	Absolute standardized differenceŦ	
Number of participants (%)		59,322 (100.00)	3,933 (6.63)	55,389 (93.36)		7,856 (100.00)	3,928 (50.00)	3,928 (50.00)		
Age group (years)										
	≤60 (%)	39,871 (67.21)	2,846 (72.36)	37,025 (66.84)	0.120Ŧ	5,626 (71.61)	2,841 (72.30)	2,785 (70.90)	0.032	
	>60 (%)	19,451 (32.80)	1,087 (27.64)	18,364 (33.16)	0.120Ŧ	2,230 (28.39)	1,087 (27.70)	1,143 (29.10)	0.032	
Sex										
	Female (%)	37,810 (63.74)	2,646 (67.28)	35,164 (63.49)	0.080	5,187 (66.03)	2,642 (67.30)	2,545 (64.80)	0.052	
	Male (%)	21,512 (36.26)	1,287 (32.72)	20,225 (36.51)	0.080	2,669 (33.97)	1,286 (32.70)	1,383 (35.20)	0.052	
Index years (Fiscal years*)										
	2011 (%)	19,532 (32.92)	686 (17.44)	18,846 (34.02)	0.386Ŧ	1399 (17.81)	686 (17.50)	713 (18.20)	0.018	
	2012 (%)	12,059 (20.33)	644 (16.37)	11,415 (20.61)	0.109	1,357 (17.27)	644 (16.4)	713 (18.2)	0.015	
	2013 (%)	9,101 (15.34)	1,111 (28.25)	7,990 (14.42)	0.342Ŧ	2,162 (27.52)	1,107 (28.2)	1,055 (26.90)	0.030	
	2014 (%)	6,148 (10.36)	508 (12.92)	5,640 (10.18)	0.086	1,055 (13.43)	507 (12.90)	548 (14.00)	0.031	
	2015 (%)	4,948 (8.34)	430 (10.93)	4,518 (8.16)	0.095	881 (11.21)	430 (10.90)	451 (11.50)	0.017	
	2016 (%)	4,186 (7.06)	312 (7.93)	3,874 (6.99)	0.036	592 (7.53)	312 (7.90)	280 (7.10)	0.031	
	2017 (%)	3,348 (5.64)	242 (6.15)	3106 (5.61)	0.023	500 (6.36)	242 (6.20)	258 (6.6)	0.017	
Charlson
Co-morbidity index (IQR)		0 (0–1)	0 (0–1)	0 (0–1)	0.219Ŧ	0 (0–1)	0 (0–1)	0 (0–1)	0.008	
Diagnoses	Diseases of esophagus, stomach, and duodenum (%)	67 (0.11)	2 (0.05)	65 (0.12)	0.023	10 (0.13)	2 (0.10)	8 (0.20)	0.043	
	Malignant neoplasms of digestive organs (%)	458 (0.77)	56 (1.42)	402 (0.73)	0.068	105 (1.34)	56 (1.40)	49 (1.20)	0.016	
	Infectious gastroenteritis (%)	440 (0.74)	22 (0.56)	418 (0.76)	0.024	61 (0.78)	22 (0.60)	39 (1.00)	0.049	
	Hypertension (%)	10,750 (18.12)	623 (15.97)	10,122 (18.27)	0.061	1,328 (16.90)	627 (16.00)	701 (17.80)	0.050	
	Diabetes mellitus (%)	5,421 (9.14)	309 (7.86)	5112 (9.23)	0.049	679 (8.64)	309 (7.90)	370 (9.40)	0.055	
	Hyperlipidemia (%)	11,883 (20.03)	618 (15.71)	11,265 (20.34)	0.121Ŧ	1,343 (17.10)	618 (15.70)	725 (18.50)	0.072	
	Renal disease (%)	2,422 (4.08)	147 (3.74)	2,275 (4.11)	0.019	361 (4.60)	147 (3.70)	214 (5.40)	0.082	
	Rheumatoid arthritis (%)	1,220 (2.06)	37 (0.94)	1,183 (2.14)	0.097	113 (1.44)	37 (0.90)	76 (1.90)	0.083	
	Hypomagnesemia (%)	172 (0.29)	1 8 (0.46)	154 (0.28)	0.030	41 (0.52)	18 (0.50)	23 (0.60)	0.018	
	Alcohol abuse (%)	333 (0.56)	42 (1.07)	291 (0.52)	0.061	78 (0.99)	42 (1.10)	36 (0.90)	0.015	
	Smoking abuse (%)	1 (0.002)	0 (0.00)	1 (0.002)	0.006	1 (0.01)	0 (0.00)	1 (0.00)	0.023	
Co-prescription	Angiotensin-converting enzyme inhibitors (ACEIs) (%)	3,427 (5.78)	191 (4.86)	3,236 (5.84)	0.044	471 (5.99)	191 (4.90)	280 (7.10)	0.096	
	Angiotensin Type II receptor antagonist (AIIA) (%)	1,408 (2.37)	75 (1.91)	1,333 (2.41)	0.034	163 (2.07)	75 (1.90)	88 (2.20)	0.023	
	Acetylsalicylic acid (aspirin) (%)	5,555 (9.36)	392 (9.97)	5,163 (9.32)	0.022	714 (9.09)	392 (10.00)	322 (8.20)	0.062	
	Non-steroidal anti-inflammatory drugs (NSAIDs) (%)	33,204 (55.97)	929 (23.62)	32,275 (58.27)	0.753Ŧ	1,953 (24.86)	929 (23.70)	1024 (26.10)	0.056	
	Steroids (%)	6,828 (11.51)	994 (25.27)	5,834 (10.53)	0.392Ŧ	1,898 (24.16)	989 (25.20)	909 (23.10)	0.048	
	Diuretics (%)	4,738 (7.99)	253 (6.43)	4,485 (8.10)	0.064	590 (7.51)	253 (6.40)	337 (8.60)	0.081	
	Statins (%)	8,696 (14.66)	528 (13.42)	8169 (14.75)	0.038	1093 (13.91)	528 (13.40)	565 (14.40)	0.027	
	Digoxin (%)	482 (0.81)	19 (0.48)	463 (0.84)	0.044	57 (0.73)	19 (0.50)	38 (1.00)	0.057	
	Aminoglycosides (%)	385 (0.65)	28 (0.71)	357 (0.64)	0.008	69 (0.88)	28 (0.70)	41 (1.00)	0.035	
Number of hospital visits (IQR)		3 (1–7)	4 (2–9)	3 (1–7)	0.157Ŧ	3 (1–7)	4 (2–9)	3 (1–7)	0.004	
Blood pressure**										
	Systolic (SD)	129.02 (19.83)	126.02 (19.11)	129.23 (19.86)	0.165Ŧ	126.67 (19.55)	126.03 (19.11)	127.29 (19.96)	0.065	
	Diastolic (SD)	76.69 (14.25)	75.91 (15.70)	76.74 (14.14)	0.055	75.99 (14.30)	75.91 (15.71)	76.07 (12.72)	0.011	
BMI (kg/m2) (SD)	Number of participants (%) = 9,123 (15.38)	24.98 (4.76)	26.21 (5.93)	24.73 (4.45)	0.281Ŧ	25.63 (5.56)	26.21 (5.94)	24.51 (4.52)	0.322Ŧ	
Notes:

PPIs, proton pump inhibitors; H2RAs, Histamine-2-receptor antagonists; IQR, Interquartile Range; SD, Standard Deviation; BMI, Body Mass Index

Ŧ Standardized difference = difference in means or proportions divided by standard error; imbalanced defined as absolute value greater than 0.10

* Thai Fiscal year: October 1st -September 30th

** Imputed blood pressure by sex and age from n (%) = 51,948 (87.57) vs 51,943 (87.56) for systolic and diastolic blood pressure, respectively.

The strength of the association between PPIs and primary outcomes was assessed by using the stratified Cox proportional hazards (CPH) model. Propensity score matching (PSM) was applied to estimate the treatment effect of PPIs, accounting for confounding by selected covariates. The propensity score was estimated using 1:1 nearest neighbor matching without replacement (Zhao et al., 2021) of PPIs with covariates. Further details of PSM were addressed in supplemental (Figs. S2, S3). To balance covariates between PPIs and H2RAs users, PSM was applied and modeled for major survival analyses.

Subgroup and sensitivity analysis

To assess whether the hazard ratio (HR) would differ by covariates, subgroup analyses were conducted with the following categorical variables: baseline age, individual medication persistent use, concomitant ASA, and types of PPIs (omeprazole vs. non-omeprazole). To address individual medication persistent use, we applied the “Medication Possession Ratio (MPR)”, which was defined as a ratio between the days of medication supply of all prescriptions filled within a time interval (Sperber, Samarasinghe & Lomax, 2017). AdhereR package in R was used to calculate MPR (Dima & Dediu, 2017) (Fig. S4).

For sensitivity analysis of unmeasured confounders (smoking and alcohol), we assumed the association strength between these confounders and each outcome based on previous studies and performed an array-based approach. This measured the impact of the prevalence range of unmeasured confounders in PPIs and H2RAs for the unknown relationship (D’Agostino McGowan, 2022; Schneeweiss, 2006). In addition, we also assessed the impact of potential joint of selection bias, misclassification of exposure, and related unmeasured confounders affecting each outcome using E-values (The Comprehensive R Archive Network, 2022). R software (version 4.0.5) was used for data management and analysis. Statistical significance was set at an alpha value of 0.05.

Results

In the fiscal year 2011, 157,522 participants aged over 20 years were included in the study cohort. However, some participants were not eligible because of neither use of PPIs or H2RAs, incomplete sex, and date of birth, whereas, some participants were excluded because study outcomes occurred during the entry period to the index date. After exclusion, there were 59,322 participants left to follow up after their individual index dates (Fig. 2).

The baseline and sociodemographic characteristics of eligible participants are presented in Table 1. The proportion of PPI and H2RAs users aged >60 years was 33 and 27 percent, respectively. In addition, the percentage of PPI and H2RAs female users were mostly observed (63 and 67 percent, respectively). Of all participants, there were only 0.11% were diagnosed with diseases of the esophagus, stomach, and duodenum using PPIs and H2RAs. PPIs users who were diagnosed with hypertension at the baseline had a higher percentage than those who used H2RAs. Less than 1% of all participants were diagnosed with smoking and alcohol abuse. The frequency of hospital visits was higher among H2RAs users than among PPIs users (median (IQR): 4 (2–9) vs 3 (1,7), respectively), whereas, the average blood pressure among all participants was less than 130 mmHg and 80 mmHg for systolic and diastolic blood pressure, respectively (Table 1).

The number of primary outcomes (CVD, IS, and PVD) and the crude incidence rates of each event are shown in Table 2. The median follow-up time of each outcome was 0.25 years before and after PSM. Compared to H2RAs, the crude incidence rates of all study outcomes were higher among PPI users (CVD: 5.41 vs 8.28, IS: 1.22 vs 3.51, and PVD: 0.09 vs 0.89, for H2RAs vs PPIs).

Table 2 Number of events, follow-up and incidence rate for the study mediators and outcomes between PPI and H2RA users (grace period = 90 days).

Measures	Before propensity score matching (N = 59,322)	After 1:1 propensity score matching (N = 7,448)	
H2RAs	PPIs	H2RAs	PPIs	
Total number of participants (%)	3,933 (6.63)	55,389 (93.36)	3,928 (50.00)	3,928 (50.00)	
Total follow-up (person-years)					
CVD	2,790.16	25,720.23	2,783.98	1,849.94	
IS	2,376.46	20,596.56	2,370.29	1,461.47	
PVD	2,293.89	18,529.93	2,287.72	1,322.12	
Median follow-up years (IQR)					
CVD	0.25 (0–0.25)	0.25 (0–0.25)	0.25 (0–0.25)	0.25 (0–0.25)	
IS	0.25 (0–0.25)	0.25 (0–0.25)	0.25 (0–0.25)	0.25 (0–0.25)	
PVD	0.25 (0–0.25)	0.25 (0–0.25)	0.25 (0–0.25)	0.25 (0–0.25)	
No. of participants with events					
CVD	151	2,130	151	163	
IS	29	722	29	48	
PVD	2	165	2	13	
Incidence rate per 100 person-years (95% CI)					
CVD	5.41 (4.58–6.35)	8.28 (7.93–8.61)	5.42 (4.59–6.36)	8.81 (7.51–10.27)	
IS	1.22 (0.82–1.75)	3.51 (3.25–3.77)	1.223 (0.82–1.76)	3.28 (2.42–4.36)	
PVD	0.09 (0.01–0.32)	0.89 (0.76–1.04)	0.09 (0.01–0.32)	0.98 (0.52–1.68)	
Note:

PPIs, proton pump inhibitors; H2RAs, histamine-2-receptor antagonists; IQR, interquartile range; CVD, cardiovascular disease; IS, ischemic stroke; PVD, peripheral vascular disease.

The strength of the association between PPIs and outcomes

Stratified Cox-PH adjusted for baseline characteristics was calculated using different models. The HRs for PPI users, as compared with H2RA users, were 1.16 (95% CI = [0.94–1.42]), 1.92 (95% CI = [1.24–2.97]), and 5.49 (95% CI = [1.22–24.61]), for CVD, IS, and PVD events, respectively. HRs based on 1:1 PSM were 1.76 (95% CI = [1.40–2.20]), 3.53 (95% CI = [2.21–5.64]), and 17.07 (95% CI = [13.82–76.25]) for CVD, IS, and PVD events, respectively (Table 3).

Table 3 Multivariable analysis of CVD, IS and PVD for PPIs and H2RAs users by unadjusted, stratified Cox-PH, and propensity score matched COX-PH models regarding grace period = 90 days.

Models	Cardio vascular diseases (CVD)1	Ischemic stroke (IS)2	Peripheral vascular diseases (PVD)3	
Crude HR [95% CI]	Adjusted HR [95% CI]	Crude HR [95% CI]	Adjusted HR [95% CI]	Crude HR [95% CI]	Adjusted HR [95% CI]	
Unadjusted	1.73 [1.47–2.04]	–	4.01 [2.77–5.82]	–	16.25 [4.03–65.62]	–	
On-treatment: stratified cox-PH adjusted for baseline covariates*	–	1.16 [0.94–1.42]	–	1.92 [1.24–2.97]	–	5.49 [1.22–24.61]	
1:1 Propensity score-matched (PSM)**	–	1.76 [1.40–2.20]	–	3.53 [2.21–5.64]	–	17.07 [13.82–76.25]	
Notes:

1* Model: adjusted for alcohol abuse, diabetes and diastolic blood pressure, stratified by clopidogrel, baseline age (≤60/>60), NSAIDs, steroids, number of hospital visits (≤2, >2), systolic blood pressure (<80/≥80), Charlson’s comorbidity index (≤1, >1) index years, and dyslipidemia.

2* Model: adjusted by clopidogrel, NSAIDs, diastolic blood pressure (<120/≥120), and alcoholic abuse, stratified by baseline age, index years, Charlson’s comorbidity index (≤1, >1), dyslipidemia, diabetes, steroids, number of hospital visits (≤2, >2), systolic blood pressure (<80/>=80).

3* Model: adjusted by number of hospital visits (≤2,>2), systolic blood pressure (<80/≥80), steroids, diastolic blood pressure (<120/≥120), and alcoholic abuse, stratified by baseline age (≤60/>60), index years, Charlson’s comorbidity index (≤1, >1), dyslipidemia, diabstes, clopidogrel, NSAIDs.

1–3** Model: Propensity score matching by sex, baseline age (≤60/>60), index years, diabetes, dyslipidemia, rheumatoid arthritis, Charlson’s comorbidity index (CCI), NSAID, Clopidogrel, Steroids, Diuretics, number of hospital visits, baseline systolic blood pressure and diastolic blood pressure.

Sub-group and sensitivity analysis

The strength of association between PPIs and each outcome was significant about medication persistence ratios >50%. The effects of concomitant use of aspirin with PPIs associated with CVD were statistically significant in only the PSM model 0.68, 95% CI = [0.49–0.94]. According to the IS outcome, for instance, the single use of PPIs associated with IS was observed in both Stratified Cox-PH and PSM models (2.03 95% CI = [1.18–3.49] and 4.209 95% CI = [2.28–7.77], respectively). In addition, over 95% of study participants were treated with Omeprazole, which showed a strong significant association with CVD, IS, and PVD events (Table 4).

Table 4 Subgroup analysis depicting the hazard ratio (HR) with 95% CI for CVD, IS, and PVD events stratified by baseline age, medication possession ratio, ASA, and generic PPIs use with a grace period of 90 days.

Subgroups	Cardio vascular diseases (CVD)	Ischemic stroke (IS)	Peripheral vascular diseases (PVD)	
Before PSM1*	After PSM1**	Before PSM2*	After PSM2**	Before PSM3*	After PSM3**	
HR	95% CI	HR	95% CI	HR	95% CI	HR	95% CI	HR	95% CI	HR	95% CI	
Baseline age													
Non-elderly (≤60)	1.278	[0.953–1.714]	2.034Ŧ	[1.475–2.805]	1.953Ŧ	[1.047–3.642]	4.209Ŧ	[2.051–8.636]	5.146	[0.519–50.987]	6.095	[0.653–56.913]	
Elderly (>60)	1.056	[0.789-1.413]	1.434Ŧ	[1.050–1.959]	1.926Ŧ	[1.130–3.281]	2.608Ŧ	[1.403–4.849]	4.957	[0.667–36.851]	4.965	[0.665–37.043]	
Medication possession ratio													
≤50%	0.794	[0.210–2.997]	0.854	[0.323–2.257]	1.047	[0.063–17.518]	1.055	[0.190–5.846]	NA	NA	NA	NA	
>50%	1.245Ŧ	[1.016–1.527]	1.818Ŧ	[1.442–2.294]	1.959Ŧ	[1.270–3.024]	3.800Ŧ	[2.310–6.251]	4.543Ŧ	[1.023–20.166]	16.876Ŧ	[3.775–75.448]	
Concomitant use of ASA													
No	1.957Ŧ	[1.431–2.678]	3.698Ŧ	[2.631–5.197]	2.032Ŧ	[1.180–3.499]	4.209Ŧ	[2.280–7.768]	4.284	[0.965–19.025]	13.535Ŧ	[2.913–62.892]	
Yes	0.840	[0.665–1.060]	0.680Ŧ	[0.490–0.944]	1.014	[0.537–1.913]	1.014	0.458–2.245]	NA	NA	NA	NA	
Generic PPIs													
Omeprazole vs H2RAs	1.228Ŧ	[1.023–1.474]	1.754Ŧ	[1.401–2.195]	1.959Ŧ	[1.311–2.928]	3.527Ŧ	[2.206–5.640]	5.261Ŧ	[1.246–22.219]	17.075Ŧ	[3.823–76.260]	
Non-omeprazole vs H2RAs	0.735	[0.362–1.491]	3.002	[0.418–21.566]	0.546	[0.052–5.703]	NA	NA	NA	NA	NA	NA	
Notes:

NA, Not applicable due to small size of events.

1* Model: adjusted for alcohol abuse, diabetes and diastolic blood pressure, stratified by clopidogrel, baseline age (≤60/>60), NSAIDs, steroids, number of hospital visits (≤2, >2), systolic blood pressure (<80/≥80), Charlson’s comorbidity index (≤1, >1) index years, and dyslipidemia.

2* Model: adjusted by clopidogrel, NSAIDs, diastolic blood pressure (<120/≥120), and alcoholic abuse, stratified by baseline age, index years, Charlson’s comorbidity index (≤1, >1), dyslipidemia, diabetes, steroids, number of hospital visits (≤2, >2), systolic blood pressure (<80/>=80).

3* Model: adjusted by number of hospital visits (≤2, >2), systolic blood pressure (<80/≥80), steroids, diastolic blood pressure (<120/≥120), and alcoholic abuse, stratified by baseline age (≤60/>60), index years, Charlson’s comorbidity index (≤1, >1), dyslipidemia, diabstes, clopidogrel, NSAIDs.

1**–3** Model (n = 2,254; 2,254; 7,856): Propensity score matching by sex, baseline age (≤60/>60), index years, diabetes, dyslipidemia, rheumatoid arthritis, Charlson’s comorbidity index (CCI), NSAID, Clopidogrel, Steroids, Diuretics, number of hospital visits, baseline systolic blood pressure and diastolic blood pressure.

Ŧ p < 0.05.

Array-based sensitivity analysis of binary unmeasured confounders (smoking and alcohol consumption) showed a significant association between exposure and all outcomes after adjusting for smoking or alcohol consumption in the PSM models. (Figs. S5–S10) Regarding multiple-bias sensitivity analysis to IS outcome (Table S5), for instance, with an E-value of 1.15, all sensitivity parameters for each bias (selection bias, misclassification of exposure, or unmeasured confounders) would have to take on for an observed hazard ratio (1.76) to be compatible with a null hazard ratio, while, E-values of 1.09 for all sensitivity parameters of each bias would have to move 95% CI of the observed hazard ratio to include the null.

Discussion

Our study was to observe the potential association between PPIs and circulatory diseases by each composite outcome (CVD, IS, and PVD). Regarding a recent systematic review and meta-analysis of observational studies, pooled HR for the association between PPIs and myocardial infarction was 1.05 (95% CI = [0.83–1.32]), whereas, pooled HR for the association between PPIs and acute cardiovascular event was 1.05 (95% CI = [0.96–1.15]) (Nolde et al., 2022). In our study, PPIs use, as compared with H2RAs, was significantly associated with CVD for the PSM model, while, it was insignificant for the stratified Cox-PH model. In our study, CVD, as a primary outcome, included heart failure, cardiac arrhythmia, and ischemic heart disease (or myocardial infarction).

Because we applied stratified CPH to unmatched data, it was assumed that the treatment effect was the same for all strata, particularly among covariates that violated the CPH assumption. However, stratified CPH might not perform well in terms of bias and power if there was a treatment-by-stratum interaction, while, treatment effects might truly differ across strata. In addition, conversion into a categorical variable as small strata as possible made stratified CPH less efficient and then led to bias estimation (In & Lee, 2019; Mehrotra, Su & Li, 2012).

According to PSM model with CVD outcome, though, the result showed a balanced covariate between PPIs and H2RAs users, the strength of association might be underestimated due to multiple biases. (Table S5) As compared with the unmatched population, changed baseline characteristics after PSM (Table S3) might affect the strength of association. To the best of our knowledge, therefore, the positive association should be interpreted deliberately.

Regarding ischemic stroke outcome, our results showed a positive association in both the unmatched and matched populations. As compared to our study, a recent systematic review and meta-analysis (Nolde et al., 2022) from five observational studies showed that the pooled HR was 1.08 (95% CI = [0.97–1.20]). Of all observational studies, however, there were conflicting results between Nolde et al. (2021) (HR = 0.98; 95% CI = [0.89–1.08]) and Wang et al. (2017) (HR = 1.11; 95% CI = [1.02–1.21]) studies, which were both assessed with moderate for overall biases. Our study resulted a significant and consistent HRs, regardless of unmeasured confounding effects from smoking and alcohol consumption (Figs. S5, S6), as well as, multiple biases with E-values for both unmatched and matched populations (Table S5).

Alternatively, the association between PPIs and peripheral vascular disease (PVD) was explored because of the potential mechanisms proposed in in vitro and in vivo studies. Additionally, long-term use of PPIs was proposed to inhibit dimethylarginine dimethylaminohydrolase 1 (DDAH1), with the accumulation of asymmetric dimethyl arginine (ADMA), while the lowering of nitric oxide (NO) as a vasoprotective molecule led to an increase in vascular cell proliferation, platelet adhesion and aggregation, and inflammation (Ghebremariam et al., 2015; Tommasi et al., 2017). Former evidence also linked PPIs intake and cardiovascular risk by lowering NO production in endothelial cells (Nolde et al., 2021), whereas, oxidative stress causing peripheral vascular disease might be associated with endothelial dysfunction by reducing NO bioavailability (Ismaeel et al., 2020). Although this study showed a positive association between PPIs and PVD, the effect size was large due to the small number of PVD events.

In our study, over 90% of PPIs given to patients were omeprazole because it was the first drug of choice for ulcer healing and variceal bleeding, recommended by the Thai national drug list (Ministry of Public Health, 2022). As compared to esomeprazole, dexlansoprazole, and lansoprazole, 8 weeks or longer use of omeprazole was more associated with cardiovascular events (Sun et al., 2017). This was relevant to our result that participants who were more likely to persistently use of PPIs during a 1-year observation period (MPR > 50%), had more likelihood of being CVD, IS, and PVD.

The major strength of our study is that patient data from a university hospital were collected systematically, while new users (NU) and active comparators (AC) were performed. For the NU approach, the time-varying hazard could be assessed, whereas, the temporality of covariate assessment was preserved (Lund, Richardson & Sturmer, 2015). The objective of selecting AC was to mitigate confounding by indication and other unmeasured participant characteristics such as baseline health status, frailty, and assignment mechanism to treatment (Lund, Richardson & Sturmer, 2015; Yoshida, Solomon & Kim, 2015). Although H2RAs were not completely replaced by PPIs indications, this was the best active comparator we had, based on recommendations in the Thai National Drug List (Ministry of Public Health, 2022).

In addition, we recruited participants who were given either PPIs or H2RAs and not only underwent endoscopy but were also used for other reasons in real-world clinical practice, such as concomitant use of PPIs with NSAIDs to prevent potential GI irritation. Since we were concerned about adherence to medication among the participants, we applied the MPR that the more persistent the use of PPIs, the higher risk of developing circulatory diseases was observed.

However, this study had some limitations. The medians of the follow-up period were short because we applied an on-treatment scheme with the right censor due to the switch of treatments to reflect the real-world use of drugs in data analysis. In addition, we did not have information on the precise indication for PPIs’ prescription. CVD diagnosis included myocardial infarction (MI), which sometimes presented with gastrointestinal manifestations. Therefore, PPIs and H2RAs may be administered to patients who first present with MI. This would lead to protopathic bias where MI was not diagnosed, and then confirmed and recorded in electronic medical records after patients were prescribed either PPIs or H2RAs (Chui et al., 2022). According to baseline characteristics of participants, however, there were less than 1% of eligible participants who were diagnosed with gastrointestinal-related disease, while, the average baseline blood pressure among participants ranged in normal (below 130/80 mm.Hg). Having said that, there was a slight effect of the misclassification of outcomes between MI and gastrointestinal symptoms in this study.

Additionally, exposure to PPIs was identified using dispensed prescriptions, whereas, self-treatment of PPIs information was not recorded, which might lead to misclassification of exposure. On-treatment with the exclusion of the immortal period (time from the entry date to the index date) was prone to selection bias. Unmeasured confounders, particularly smoking, alcohol consumption, physical activity, and individual BMI, may affect the estimation of HR for circulatory diseases based on Array-based and E-values for multiple-bias sensitivity analysis.

Missing information on outcome measures could not be avoided because there was the loss of follow-up among patients in the study hospital while those patients were diagnosed later by other hospitals. The lack of generalizability of the study was still a limitation because an only hospital was studied. Although there is no reliability test for ICD-10, all ICD codes were directly assigned from the physician based on the patient’s conditions. Finally, we cannot exclude the power issue with H2RAs in this study, as most patients were treated with PPIs. For further research, we suggest exploring a potentially explainable mechanism of PPIs associated with circulatory diseases through mediators anconfirming the association between PPIs and PVD. In clinical practice, the risks and benefits of PPIs use, including combination with other drugs, must be weighed out, whereas, the necessity of PPIs use should be reassessed during the treatment plan of patients who are at risk of circulatory diseases.

Conclusion

In conclusion, the effects of PPIs associated with ischemic stroke and peripheral vascular disease were significant in a Cox proportional hazards and propensity score matching analysis in a Thai hospital-based cohort. This is the first study to report an association between PPIs use and PVD, meanwhile, we provided alternative results from the Asian population to compare with results from other regions.

Supplemental Information

Supplemental Information 1 Supplementary Materials.

Click here for additional data file.

Supplemental Information 2 Deidentified datasets.

Click here for additional data file.

Supplemental Information 3 Code books.

Click here for additional data file.

We thank Mr. Hsin-Jin (Jack) Huang for his assistance and advice on the wrangle data using the R program.

Additional Information and Declarations

Competing Interests

Author Contributions

Ethics

Data Availability

The authors declare that they have no competing interests.

Tanavij Pannoi conceived and designed the experiments, performed the experiments, analyzed the data, prepared figures and/or tables, and approved the final draft.

Chissanupong Promchai performed the experiments, authored or reviewed drafts of the article, and approved the final draft.

Penjamaporn Apiromruck performed the experiments, authored or reviewed drafts of the article, and approved the final draft.

Suwikran Wongpraphairot performed the experiments, authored or reviewed drafts of the article, and approved the final draft.

Yaa-Hui Dong conceived and designed the experiments, authored or reviewed drafts of the article, and approved the final draft.

Chen-Chang Yang conceived and designed the experiments, authored or reviewed drafts of the article, and approved the final draft.

Wen-Chi Pan conceived and designed the experiments, authored or reviewed drafts of the article, and approved the final draft.

The following information was supplied relating to ethical approvals (i.e., approving body and any reference numbers):

Institutional review boards of National Yang Ming Chiao Tung University (the approval number is YM108156E).

The following information was supplied regarding data availability:

The deidentified raw data are available in the Supplemental Files.

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
