# Peer review of "Risk of circulatory diseases associated with proton-pump inhibitors: a retrospective cohort study using electronic medical records in Thailand"

_PeerJ, doi:10.7717/peerj.16892_

## Round 0.1 · original submission · Minor Revisions

The manuscript was reviewed by the editor and 3 experts and some important comments were raised.

A major limitation of this paper is with the 3 months of follow-up period, which really limits the applicability of the findings.

In addition, the authors are suggested to address the issue of “how a University in Taiwan could provide IRB approval for a study undertaken in Thailand”.

Reviewer 1 ·

Basic reporting

1. Page 9, Line 68. Is the use of an in-vitro study a potential limitation in determining the association of potential mechanisms?
2. Page 10, Line 95. How does using non-PPI uses as a control lead to indication bias?
3. Page 18, Line 238. What is the PSM model in relation to this study. How exactly was it used?
4. Page 20, Line 297. How were the outcomes of MI misclassified in this study?
5. Page 21, Line 317-319. This is not the first study to report an association between PPIs and PVD. This sentence should be rephrased or removed entirely. Examples with DOIs include: https://www.mayoclinicproceedings.org/article/S0025-6196(21)00233-0/fulltext, https://doi.org/10.1016/j.mayocp.2021.02.025.

Experimental design

1. Page 12, Line 126. Is it acceptable to predefine the discontinuation of medicine if it was not already explicitly stated?

Validity of the findings

1. Page 16, Line 214. Explain how array-based sensitivity analysis was used in this study.

Additional comments

This manuscript analyzes the risk of circulatory disease (CVD) and other cardiovascular factors as an effect of proton pump inhibitors (PPI). This study is rigorous with many tables and charts to back up the information provided, and conclusions drawn.

Reviewer 2 ·

Basic reporting

The English is clear, although I suggest having a colleague proficient in English review the introduction section. The importance of this study is well justified, but some content in the introduction can be moved to the discussion section. For example, the detailed discussion of the previous findings and their limitations. I suggest covering the following discussion points in the introduction section: (1) why studying the composite outcomes rather than the three outcomes of interest separately; (2) how the results from this study will be clinically meaningful in practice; (3) whether there is existing independent research or a meta-analysis focusing on the same exposure and composite outcome.

Experimental design

The authors have thoroughly discussed the essential components of the method section.

Comment 1: I suggest formatting the visual demonstration of the study design (Figure 1) based on a generally acknowledged paper (as attached) for readability. (Schneeweiss S, Rassen JA, Brown JS, et al. Graphical Depiction of Longitudinal Study Designs in Health Care Databases. Ann Intern Med. 2019;170(6):398-406. doi:10.7326/M18-3079)

Comment 2: Please report or cite previous research about the performance of algorithms to identify exposure and outcome. It is a good practice to include detailed information in supplementary materials to ensure transparency.

Comment 3: Information about handling missing data in line 141-143 was not sufficient for reproducibility. For example, what are the reasons for missing SBP and DBP, and what missing mechanism did the authors assume accordingly? Also, please provide the number of multiply imputed datasets, as well as how the authors aggregated results from the multiply imputed datasets.

Comment 4: Line 155: Supplementary Figure 1S is assessing the assumption of proportionality, not the strength of the association. Please correct or edit it to avoid confusion.

Comment 5: Why were only a subset of covariates included in the love plot?

Comment 6: The ASD of BMI is still above 0.1. Did you control for BMI in the Cox model?

Overall, this study is well-conducted.

Validity of the findings

1. Please demonstrate results with numbers, rather than using terms such as 'higher' or 'less than'. For example, in line 186-187 and line 192.
2. The strength and limitations are very well stated in the discussion section. I would further recommend restructuring the first three paragraphs. For example, (1) summarize the findings of this study in the first paragraph; (2) the second and third paragraphs discuss technical details and limitations of the methods. This might overlap with the discussion of limitations in the following paragraphs. I suggest interpreting the results accordingly based on the techniques used and explaining how the results from this study will benefit or support clinical decision-making.

Reviewer 3 ·

Basic reporting

Professional English writing with clear statement and detailed analysis.
This article has a professional article structure, lean tables and figures.
The hypothesis is well analysed and stated according to the data.

Experimental design

The article has a well defined research question, it's closely related to clinical practice, and the retrospective study design fits the study well. The dataset with long follow up years makes the result and the conclusion convincing.
Methods and results are described with sufficient details and information.

Validity of the findings

Conclusions are well stated, linked to original research question & limited to supporting results.
This article address that we should be aware of the conenction between PPI and cardiac illnesses.

Additional comments

NA

---

## Round 0.2 · accepted · Accept

The authors have properly addressed the comments from the reviewers. The previous reviewers were invited to review the revision. This manuscript is ready for publication.